# Plant-Scale Circular Economy Using Biological Reuse of Electrolyte Residues in the Amino Acid Industry

**DOI:** 10.3390/bioengineering12010024

**Published:** 2024-12-30

**Authors:** Jun-Woo Kim

**Affiliations:** CJ BIO Research Institute, CJ CheilJedang Corp., Suwon-si 16495, Gyeonggi-do, Republic of Korea; junwoo.kim1@cj.net

**Keywords:** amino acid, circular economy, electrolyte, fermentation

## Abstract

The amino acid industry generates significant amounts of electrolyte residues, such as ammonium sulfate, acetic acid, and phosphoric acid, which cause challenges to sustainability. This short article investigates the feasibility of implementing a plant-scale circular economy through the recycling and biological reuse of these electrolyte residues. Scenario analyses of L-lysine (LYS) HCl, L-methionine (MET), and L-cysteine (CYS) HCl production highlight the environmental and economic benefits of the plant-scale circular economy. Despite these advances, challenges remain, including improving reuse rates for residues and reducing sodium ion content in the salt. This analysis emphasizes the need for integrated process design to enhance the circular economy, not only in amino acid production but also in other fermentation-based industries.

## 1. Introduction

Amino acids are the most extensively produced products via fermentation [1], following alcohols [2]. The global annual production of amino acids is estimated to be approximately 10 million metric tons [1]. L-glutamic acid, used as a flavor enhancer, is reported to have an annual production volume of around 4,000,000 metric tons [3]. L-lysine (LYS), used as a feed additive, is reported to be produced at an annual volume of approximately 30,000,000 metric tons [4]. Although methionine (MET) is primarily produced in DL-form derived from fossil fuels, with an annual production volume of about 20,000,000 metric tons [5], it can also be produced in the L-form through fermentation processes [6,7]. The largest L-methionine plant (CJ CheilJedang Co.) is located in Kerteh, Malaysia, with a production capacity of 180,000 metric tons per year. Additionally, L-threonine, L-tryptophan, L-valine, L-isoleucine, L-glutamine, L-leucine, L-arginine, L-histidine, L-alanine, L-proline, L-serine, and L-tyrosine are industrially produced using fermentation processes [8,9].

The production of amino acids can generate large amounts of electrolyte residues composed of substances such as ammonia, sulfuric acid, acetic acid, and phosphoric acid. These electrolyte residues are considered waste, hindering the sustainability of amino acid manufacturing. This short article introduces the concept of a circular economy on a plant scale in fermentation-based amino acid production processes. Achieving a circular economy is the most fundamental solution to managing residues [10,11]. To realize this, an integrated process design that enables the reuse of electrolyte residues in fermentation processes must be developed during the stages of product development and plant engineering [12]. The circular economy of electrolyte residues can contribute not only to environmental sustainability but also to improving economic value. Here, we aim to present examples and impacts of electrolyte circular economy through scenario analysis in productions of LYS HCl, MET, and L-cysteine (CYS) HCl.

## 2. Scenario Analysis

In this article, three commercially significant amino acids were selected as example products for the scenario analysis. LYS is typically supplied in the form of HCl crystalline salt powder. LYS supports animal growth rate [13] and calcium absorption [14], acting as the primary growth factor for pigs [9]. The increased use of amino acids as feed additives reduces the reliance on crude protein, playing a crucial role in lowering CO_2_ emissions in the livestock industry [15,16]. MET is generally supplied as a pure crystalline powder. MET contributes to animal growth rate [17], gut health [18], immune resistance [19], and antioxidant activity [20], serving as the primary growth factor for chickens [9]. CYS is typically supplied in the form of HCl crystalline salt powder. CYS can be used as a food additive to provide chicken- and beef-like flavors [21]. Waste reduction and economic value of the electrolyte circular economy in the production of LYS HCl, MET, and CYS HCl were analyzed.

## 3. Ammonium Sulfate Residue in LYS HCl Production

Figure 1 illustrates the production process of LYS HCl [22,23]. Ion-exchange is the most significant separation and purification unit operation in the production of LYS HCl. In the ion-exchange process, LYS adsorbed onto the cation-exchange resin is eluted with an excess of ammonium hydroxide aqueous solution. Meanwhile, the raffinate is neutralized with sulfuric acid, resulting in ammonium sulfate (AMS) solution residues. The entire AMS solution would need to be discharged as waste in the form of fertilizer without an appropriate recycling and reuse process. However, ammonia and sulfuric acid can be utilized as a nitrogen source and pH buffering agent, respectively, in fermentation. Therefore, industrial processes have developed a method to recover pure AMS from the raffinate through evaporative crystallization, allowing it to be reused in fermentation [24].

Through AMS recycling, a 71% reduction in waste AMS can be achieved. Additionally, as shown in Table 1, a reduction in raw materials such as NH_3_ and H_2_SO_4_ can lead to a unit cost saving of 88 USD/t. A steam cost increase of 48 USD/t is calculated with a case applying a five-effect evaporator and single-effect evaporative crystallizer for AMS production. In total, a variable cost reduction of 40 USD/t can be achieved. The production cost of L-lysine is influenced by various factors, including the price of the carbon source, but it can be assumed to be approximately 2000 USD/t in practice. The recovery and reuse of AMS is expected to reduce the production cost of L-lysine by about 2%. Assuming a global LYS production volume of 3,000,000 t/y, the maximum expected benefit is 120,000,000 USD/y. Nonetheless, a limitation remains as AMS mother liquor is still not recycled and must be discharged. Although AMS mother liquor can be used as fertilizer to achieve a macro-scale circular economy, utilizing it in fermentation processes would be more efficient in terms of both environmental and cost considerations. AMS mother liquor is supplied to fertilizer companies or nearby large farms at a low price. It is generally used as a raw material for liquid or composite solid fertilizers, diluted or concentrated to an appropriate composition based on the condition of the soil and crops.

## 4. Acetic Acid Residue in L-Methionine Production

Figure 2 illustrates the production process of MET [6]. Through fermentation, O-acetyl-L-homoserine (AH) is first produced, followed by the enzymatic conversion reaction to produce MET. During this process, acetic acid is generated as a byproduct in a molar ratio equivalent to MET. Acetic acid is emitted in the vapor phase during the evaporative crystallization processes of MET and AMS. The concentration of acetic acid in the vapor phase varies depending on the composition of the process liquid, pH, and evaporation process conditions [7]. Aqueous solution of acetic acid generated through condensation is treated with NH_3_, and then ammonium acetate (AMAC) aqueous solution is formed. AMAC aqueous solution is concentrated via reverse osmosis (RO) and evaporation processes. This enables the reuse of acetic acid in the fermentation process [25]. AH can be synthesized from L-homoserine-derived sugar and acetyl-coenzyme A derived from acetic acid, so the recovered acetic acid can be directly used as a raw material in the fermentation process [6].

Through AMAC recycling and reuse, an 87% reduction in acetic acid waste can be achieved. Additionally, as shown in Table 2, a reduction in sugar, the major raw material, can result in a unit cost saving of 170 USD/t. A steam cost increase of 48 USD/t is calculated based on a case where reverse osmosis and a six-effect evaporator are applied for AMAC production. In total, a variable cost reduction of 122 USD/t can be achieved. Assuming a global L-MET production volume of 180,000 t/y, the maximum expected benefit is 22,000,000 USD/y. Nevertheless, there remain limitations as a considerable amount of AMS and acetic acid are still discharged. However, AMS and acetic acid can be used as fertilizers and plant growth promoters to achieve a macro-scale circular economy. Utilizing them in fermentation processes would be more efficient in terms of both environmental and cost considerations. The liquid fertilizer containing acetic acid and ammonia from L-methionine can be used both as a nitrogen fertilizer and herbicide. The L-methionine plant is located in Malaysia, and therefore, liquid fertilizer is sold for the purpose of preventing Ganoderma disease in the surrounding palm oil plantations and supplying nitrogen at a low price [6].

## 5. Phosphoric Acid Residue in L-Cysteine Production

Figure 3 illustrates the production process of CYS HCl [26]. Through fermentation, O-phospho-L-serine (OPS) is first produced, followed by an enzymatic conversion reaction to produce CYS. During this process, phosphoric acid is generated as a byproduct in a molar ratio equivalent to CYS. In the simulated moving bed (SMB) process, CYS is discharged as the extract, while phosphoric acid is discharged as the raffinate. The raffinate is then recovered in the form of hydrogen phosphate (HPO_4_^2–^) salt through an evaporative crystallization process. Depending on the process conditions, the salt may have varying compositions of sodium and ammonium ions. Ammonium ions can be used as a nitrogen source in fermentation, but sodium ions cannot be utilized as active components in fermentation. This implies that minimizing the use of NaOH is essential for ensuring high economic efficiency and environmental sustainability.

The hydrogen phosphate salt recovery process was initially developed in a sodium ion-rich form [27]. Recently, process improvement involving the addition of ammonia during the raffinate make-up stage has enabled an increase in the ammonium ion content within the salt [28]. A future area of research is replacing NaOH with NH_3_ in the enzymatic conversion reaction. The use of NH_3_ may increase the vaporization tendency of the sulfur source, so the physical property, hazard, and safety of the new electrolyte system need to be reassessed. Ammonium ions may impact the enzyme conversion process, requiring process optimization and potentially the development of new enzymes. Furthermore, due to changes in the enzyme conversion reactant, the processes for L-cysteine and phosphate salt crystal production may need to be redesigned. The increased use of ammonia is likely to improve the recovery and reuse rate of phosphoric acid, which may require new design considerations for the fermentation process conditions and strains. These would provide both economic and environmental values by increasing the ammonium ion content in the recycled hydrogen phosphate salt.

## 6. Application in Other Industries

Beyond the examples presented in the scenario analysis, integrated process design perspectives are necessary for the production processes of other bioproducts as well. For instance, in the production of organic acids, such as succinic acid, acetic acid, and formic acid, the generation of electrolyte residues based on the choice of counter ions can have significant environmental and economic impacts. For example, recovering succinic acid in the form of calcium salts may optimize production costs [29]. However, a more detailed analysis and process design for recycling calcium salts generated during subsequent applications requires further in-depth research.

Diamine can serve as a raw material for bio-nylon [2]. It potentially has a significant impact on the future bioeconomy. However, handling diamine inevitably generates counter ions, which may lead to the discharge of electrolytes. In such cases, the focus should not be solely on the production cost and environmental performance of diamine. Instead, the process must be designed with an emphasis on the management and treatment of electrolyte residues, considering both integrated processes and application perspectives.

L-citrulline, one of the dietary supplements, can be produced through an enzymatic conversion of L-arginine HCl, generating ammonium chloride as a byproduct. Because ammonium chloride dissolves in aqueous alcohol solutions, it remains in the mother liquor during the subsequent antisolvent crystallization process. However, chloride ions are highly corrosive, which can induce issues when reusing them in arginine fermentation. Therefore, conducting the enzymatic conversion of L-citrulline using L-arginine free form and selecting an appropriate counter ion is a significant research topic. Additionally, designing a cost-effective downstream process that considers the type of electrolytes generated is necessary.

## 7. Conclusions

In summary, this short article demonstrates the following: (1) Through scenario analysis of commercial processes for LYS, MET, and CYS, recycling and biological reuse of electrolyte residues in amino acid processes are feasible. (2) Such a plant-based circular economy system provides both a decrease in electrolyte residue emissions and an increase in economic benefits. (3) However, several challenges remain. Improvement of the reuse rate of AMS residues generated during the production of LYS HCl and MET is required. Additionally, increasing the proportion of ammonium ions in the hydrogen phosphate salt generated during the production of CYS HCl is another task to address. (4) The concept of the electrolyte circular economy should be expanded and applied to other categories of fermentation-based products.

## Figures and Tables

**Figure 1 bioengineering-12-00024-f001:**
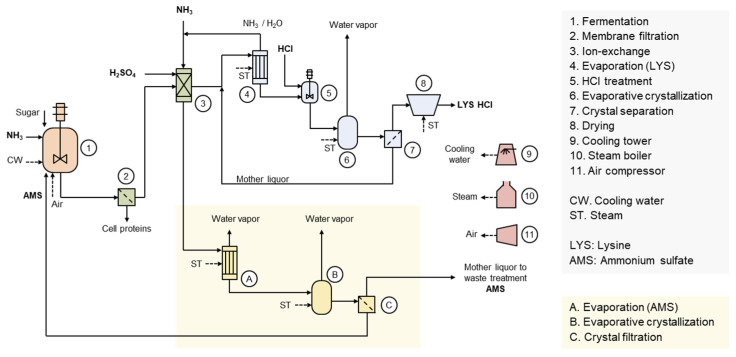
Production process of L-lysine HCl with ammonium sulfate recycling and reuse. L-lysine sulfate dissolved in the broth is produced through fermentation. The cells in the broth are removed by membrane filtration, and the permeate is transferred to the ion-exchange process. L-lysine adsorbed onto the cation exchange resin is eluted with ammonia solution and then concentrated through evaporation, during which ammonia is removed. After the addition of hydrochloric acid, the product is obtained as high-purity L-lysine HCl through evaporative crystallization, crystal separation, and drying processes. AMS solution, discharged as raffinate from the cation exchange process, undergoes evaporation, evaporative crystallization, and crystal separation to obtain high-purity AMS crystals. The high-purity AMS can be reused in the L-lysine fermentation.

**Figure 2 bioengineering-12-00024-f002:**
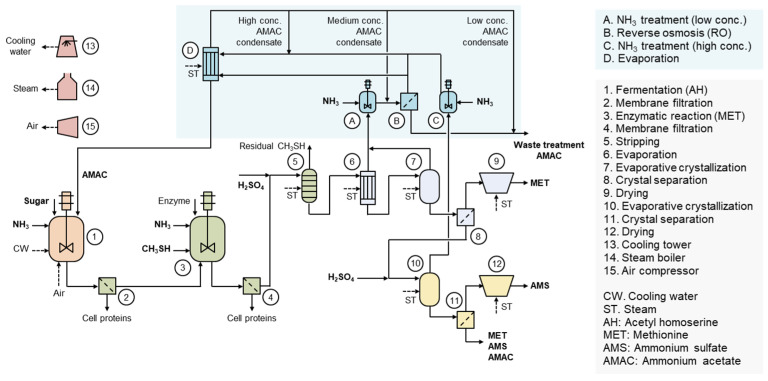
Production process of L-methionine with acetic acid recycling and reuse. AH is first produced through fermentation, and the cells are removed via the membrane process. Subsequently, an enzymatic reaction between AH and CH_3_SH generates equimolar amounts of L-methionine and acetic acid. The enzyme reaction mixture undergoes the membrane process to remove enzymes and residual cells, and CH_3_SH is removed through the stripping process. L-methionine product is then produced through evaporation, evaporative crystallization, crystal separation, and drying. The condensate generated during the evaporation and evaporative crystallization processes contains acetic acid, which is converted into the AMAC solution through ammonia treatment. The low-concentration AMAC solution is concentrated via reverse osmosis, while the high-concentration AMAC solution is concentrated through evaporation. The final concentrated AMAC solution can then be reused in the AH fermentation.

**Figure 3 bioengineering-12-00024-f003:**
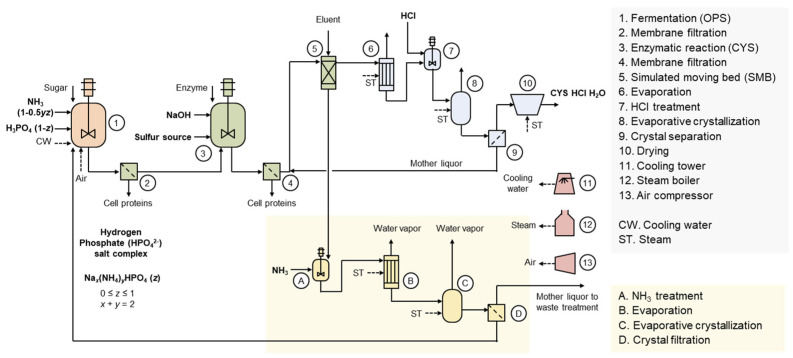
Production process of L-cysteine HCl with phosphoric acid recycling and reuse. OPS is first produced through fermentation, and the cells are removed via the membrane process. Subsequently, an enzymatic reaction between OPS and the sulfur source generates equimolar amounts of L-cysteine and phosphoric acid. The enzyme reaction mixture undergoes the membrane process to remove enzymes and residual cells. The L-cysteine is then discharged as the extract through the simulated moving bed process, while phosphoric acid is discharged as the raffinate. The extract undergoes hydrochloric acid addition, followed by evaporative crystallization, crystal separation, and drying for the production of L-cysteine HCl monohydrate product. The raffinate undergoes ammonia treatment, evaporative crystallization, and crystal separation for the production of sodium ammonium phosphate crystals. Sodium ammonium phosphate can be reused in OPS fermentation. The amount of sodium added in the previous step affects the sodium content in the sodium ammonium phosphate crystals.

**Table 1 bioengineering-12-00024-t001:** Economic impact analysis of AMS reuse in LYS HCl production. The price information for NH_3_, H_2_SO_4_, and steam is based on the average local prices in Indonesia for 2023.

Item	Δ Unit Consumption (t/t)	Price (USD/t)	Δ Unit Cost (USD/t)
NH_3_	–0.093	500	–47
H_2_SO_4_	–0.27	150	–41
Steam	1.6	30	48
Variable cost			–40

**Table 2 bioengineering-12-00024-t002:** Economic impact analysis of acetic acid reuse in MET production. The price information for sugar and steam is based on the average local prices in Malaysia for 2023.

Item	Δ Unit Consumption (t/t)	Price (USD/t)	Δ Unit Cost (USD/t)
Sugar	–0.34	500	–170
Steam	1.6	30	48
Variable cost			–122

## Data Availability

No new data were created or analyzed in this study.

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
