# Peer review of "Plant-Scale Circular Economy Using Biological Reuse of Electrolyte Residues in the Amino Acid Industry"

_bioengineering, 2024, doi:10.3390/bioengineering12010024_

Round 1
Reviewer 1 Report
Comments and Suggestions for Authors
Plant-Scale Circular Economy using Biological Reuse of Electrolyte Residues in Amino Acid Industry
The author Jun-Woo Kim presents examples and the benefits of an electrolyte circular economy through scenario analyses in the production of LYS HCl, MET, and L-cysteine (CYS) HCl. The amino acid industry produces notable amounts of electrolyte byproducts, such as ammonium sulfate, acetic acid, and phosphoric acid, which pose sustainability challenges. This article explores the implementation of a plant-scale circular economy by focusing on the recycling and biological reuse of these electrolyte byproducts. The scenario evaluations of L-lysine (LYS) HCl, L-methionine (MET), and L-cysteine (CYS) HCl production emphasize the significant environmental and economic advantages that can be achieved through a plant-scale circular economy.
The author is suggested to address and provide an appropriate response to the comments.
· Although AMS mother liquor can be used as fertilizer to achieve a macro-scale circular economy, utilizing it in fermentation processes would be more efficient in terms of both environmental and cost considerations--- It is acknowledged that AMS mother liquor may be used as fertilizer; however, can this liquor be applied directly? What are the necessary processes and preparations for using the AMS for fertilizer application?
· Aqueous solution of acetic acid generated through condensation is 91 treated with NH3, and then ammonium acetate (AMAC) aqueous solution is formed. AMAC aqueous solution is concentrated via reverse osmosis (RO) and evaporation processes. This enables the reuse of acetic acid in the fermentation process--- What is the role of acetic acid in the fermentation process? What kind of fermentation process necessitates acetic acid?
· Through AMAC recycling and reuse, an 87% reduction in acetic acid waste can be achieved. Additionally, as shown in Table 2, a reduction in sugar, the major raw material, can result in a unit cost saving of 170 USD/t--- What methods were employed to achieve an 87% value for AMAC recycling and reuse, resulting in an 87% reduction in acetic acid waste?
· Although AMS and acetic acid can be used as fertilizers and plant growth promoters to achieve a macro-scale circular economy. Utilizing them in fermentation processes would be more efficient in terms of both environmental and cost considerations—in what way can acetic acid be used as fertilizer?
Author Response
Detailed Response to Reviewers
Reviewer #1
Dear reviewer,
Thank you for your feedback. I appreciate your comments for improving this manuscript. Following your recommendations, I extended discussion and tried to provide more considerable goals. Please find our detailed answers to your comments below in red.
Plant-Scale Circular Economy using Biological Reuse of Electrolyte Residues in Amino Acid Industry
The author Jun-Woo Kim presents examples and the benefits of an electrolyte circular economy through scenario analyses in the production of LYS HCl, MET, and L-cysteine (CYS) HCl. The amino acid industry produces notable amounts of electrolyte byproducts, such as ammonium sulfate, acetic acid, and phosphoric acid, which pose sustainability challenges. This article explores the implementation of a plant-scale circular economy by focusing on the recycling and biological reuse of these electrolyte byproducts. The scenario evaluations of L-lysine (LYS) HCl, L-methionine (MET), and L-cysteine (CYS) HCl production emphasize the significant environmental and economic advantages that can be achieved through a plant-scale circular economy.
The author is suggested to address and provide an appropriate response to the comments.
Comment 1: Although AMS mother liquor can be used as fertilizer to achieve a macro-scale circular economy, utilizing it in fermentation processes would be more efficient in terms of both environmental and cost considerations--- It is acknowledged that AMS mother liquor may be used as fertilizer; however, can this liquor be applied directly? What are the necessary processes and preparations for using the AMS for fertilizer application?
Response 1:
It is not usually used directly. I have added following sentences (Line 90–93):
“AMS mother liquor is supplied to fertilizer companies or nearby large farms at a low price. It is generally used as a raw material for liquid or composite solid fertilizers, diluted or concentrated to an appropriate composition based on the condition of the soil and crops.”
Comment 2: Aqueous solution of acetic acid generated through condensation is 91 treated with NH3, and then ammonium acetate (AMAC) aqueous solution is formed. AMAC aqueous solution is concentrated via reverse osmosis (RO) and evaporation processes. This enables the reuse of acetic acid in the fermentation process--- What is the role of acetic acid in the fermentation process? What kind of fermentation process necessitates acetic acid?
Response 2:
Acetic acid is used in fermentation merely as a raw material for O-acetyl-L-homoserine. I have added the following sentences (Line: 106–109):
“AH can be synthesized from L-homoserine derived sugar and acetyl-coenzyme A derived from acetic acid, so the recovered acetic acid can be directly used as a raw material in the fermentation process [6].”
Comment 3: Through AMAC recycling and reuse, an 87% reduction in acetic acid waste can be achieved. Additionally, as shown in Table 2, a reduction in sugar, the major raw material, can result in a unit cost saving of 170 USD/t--- What methods were employed to achieve an 87% value for AMAC recycling and reuse, resulting in an 87% reduction in acetic acid waste?
Response 3:
I have already explained the AMAC recycling and reuse method along with the reference (Line 103–106).
Comment 4: Although AMS and acetic acid can be used as fertilizers and plant growth promoters to achieve a macro-scale circular economy. Utilizing them in fermentation processes would be more efficient in terms of both environmental and cost considerations—in what way can acetic acid be used as fertilizer?
Response 4:
CJ BIO’s L-methionine plant is located in Malaysia, so it is primarily used as a liquid fertilizer for palm oil plantation. I have added the following sentences (Line 132–136):
“The liquid fertilizer containing acetic acid and ammonia from L-methionine can be used both as a nitrogen fertilizer and herbicide. L-Methionine plant is located in Malaysia, and therefore, liquid fertilizer is sold for the purpose of preventing Ganoderma disease in the surrounding palm oil plantations and supplying nitrogen with a low price [6].

Reviewer 2 Report
Comments and Suggestions for Authors
After carefully reviewing the manuscript titled 'Plant-Scale Circular Economy using Biological Reuse of Electrolyte Residues in the Amino Acid Industry,' I believe the work has significant potential. However, to enhance its clarity, depth, and overall quality, I recommend that there be major revisions."
-The introduction provides a solid overview of the importance of amino acid production via fermentation and its environmental impact due to large amounts of electrolyte residues. However, a brief mention of specific global challenges or statistics regarding waste disposal in the industry would further strengthen the urgency of adopting circular economy practices.
The schematic in Figure 1 effectively communicates the circular economy concept for electrolyte residue reuse in amino acid production. However, adding more specific labels or steps in the process would be helpful to make it clearer to readers unfamiliar with the technical details.
-Table 1 demonstrates the potential cost savings associated with ammonium sulfate recycling. In contrast, it’s evident that recycling results in cost reduction, the table would benefit from including a comparison of these costs to the overall cost of production to provide context for the impact on the bottom line.
-The scenario analysis for LYS, MET, and CYS HCl production is well-executed and provides useful insights. However, more detailed information on how different production scales affect the economic feasibility of implementing the circular economy model would make the findings more applicable to various plant sizes.
-The process diagram in Figure 2 for LYS HCl production with ammonium sulfate recycling is clear, but it would benefit from a brief explanation of each step in the caption. This would make the figure more informative, especially for readers who might not have a background in the field.
-The process flow for phosphoric acid recycling in the production of CYS HCl is an innovative approach. However, more emphasis should be placed on the practical challenges of scaling up this process. A section discussing potential obstacles in industrial applications would be valuable.
-The paper addresses the environmental benefits of the circular economy model. However, it would be useful to include a more detailed life cycle assessment (LCA) to quantify these environmental benefits, such as reductions in CO2 emissions or water usage.
-The analysis provided in Table 2 is comprehensive, especially in terms of raw material and energy savings. However, the economic impact of different market conditions (such as fluctuating raw material prices) could be explored to make the analysis more robust and applicable to different regions or industries.
-The paper outlines the economic advantages of recycling electrolyte residues, but a deeper discussion of the long-term sustainability of such systems, including potential regulatory challenges or future trends in waste management regulations, would add value to the argument.
-While the paper mentions that further research is required, such as increasing ammonium ion content in phosphoric acid salts, a more detailed roadmap or timeline for such research would make the paper’s conclusion more actionable. Highlighting specific areas where breakthroughs are needed (e.g., material innovations or process engineering) would guide future work in the field.
Author Response
Detailed Response to Reviewers
Reviewer #2
Dear reviewer,
Thank you for your feedback. I appreciate your comments for improving this manuscript. Following your recommendations, I extended discussion and tried to provide more considerable goals. Please find our detailed answers to your comments below in red.
After carefully reviewing the manuscript titled 'Plant-Scale Circular Economy using Biological Reuse of Electrolyte Residues in the Amino Acid Industry,' I believe the work has significant potential. However, to enhance its clarity, depth, and overall quality, I recommend that there be major revisions."
Comment 1: The introduction provides a solid overview of the importance of amino acid production via fermentation and its environmental impact due to large amounts of electrolyte residues. However, a brief mention of specific global challenges or statistics regarding waste disposal in the industry would further strengthen the urgency of adopting circular economy practices.
Response 1:
Your comment is absolutely correct. However, I was unable to find specific statistics or quantitative data on the amount of waste generated in the amino acid industry. Although I was not able to present the direct statistical data, I believe that this opinion article itself will highlight the importance of electrolyte waste recycling and the adoption of the circular economy in the amino acid industry. Additionally, I believe that the mention of the economic scales in amino acid industry (Line 19–31) and practical economic effects (Line 85–87 and 126–128) provides supporting evidence.
Comment 2: The schematic in Figure 1 effectively communicates the circular economy concept for electrolyte residue reuse in amino acid production. However, adding more specific labels or steps in the process would be helpful to make it clearer to readers unfamiliar with the technical details.
Response 2:
I believe that Figure 1 is more suitable as a graphical abstract rather than as part of the manuscript. Instead, the specific description has been added to the captions of other figures that depict the process flow diagram.
Comment 3: Table 1 demonstrates the potential cost savings associated with ammonium sulfate recycling. In contrast, it’s evident that recycling results in cost reduction, the table would benefit from including a comparison of these costs to the overall cost of production to provide context for the impact on the bottom line.
Response 3:
The production cost of L-lysine is generally around 2 000 USD/t, I have added the following sentences (Line 82–85):
“The production cost of L-lysine is influenced by various factors, including the price of the carbon source, but it can be assumed to be approximately 2 000 USD/t in practice. The recovery and reuse of AMS is expected to reduce the production cost of L-lysine by about 2%.”
Comment 4: The scenario analysis for LYS, MET, and CYS HCl production is well-executed and provides useful insights. However, more detailed information on how different production scales affect the economic feasibility of implementing the circular economy model would make the findings more applicable to various plant sizes.
Response 4:
The impact of this study is primarily related to variable costs associated with NH3, H2SO4, steam, and sugar. Unlike fixed costs, variable costs are not affected by plant size. However, as the scale increases, the overall economic impact grows proportionally. These aspects have already been mentioned in the manuscript (Line 85–87 and 126–128)
Comment 5: The process diagram in Figure 2 for LYS HCl production with ammonium sulfate recycling is clear, but it would benefit from a brief explanation of each step in the caption. This would make the figure more informative, especially for readers who might not have a background in the field.
Response 5:
I have added the following specific details about the process flow diagram to the captions of all figures (Line 69–77, 111–121, and 152–163):
“Figure 1. Production process of L-lysine HCl with ammonium sulfate recycling and reuse. L-Lysine sulfate dissolved in the broth is produced through fermentation. The cells in the broth are removed by membrane filtration, and the permeate is transferred to the ion-exchange process. L-Lysine adsorbed onto the cation exchange resin is eluted with ammonia solution and then concentrated through evaporation, during which ammonia is removed. After the addition of hydrochloric acid, the product is obtained as high-purity L-lysine HCl through evaporative crystallization, crystal separation, and drying processes. AMS solution, discharged as raffinate from the cation exchange process, undergoes evaporation, evaporative crystallization, and crystal separation to obtain high-purity AMS crystals. The high-purity AMS can be reused in the L-lysine fermentation.”
“Figure 2. Production process of L-methionine with acetic acid recycling and reuse. AH is first produced through fermentation, and the cells are removed via the membrane process. Subsequently, an enzymatic reaction between AH and CH3SH generates equimolar amounts of L-methionine and acetic acid. The enzyme reaction mixture undergoes the membrane process to remove enzymes and residual cells, and CH3SH is removed through the stripping process. L-Methionine product is then produced through evaporation, evaporative crystallization, crystal separation, and drying. The condensate generated during the evaporation and evaporative crystallization processes contains acetic acid, which is converted into the AMAC solution through ammonia treatment. The low-concentration AMAC solution is concentrated via reverse osmosis, while the high-concentration AMAC solution is concentrated through evaporation. The final concentrated AMAC solution can then be reused in the AH fermentation.”
“Figure 3. Production process of L-cysteine HCl with phosphoric acid recycling and reuse. OPS is first produced through fermentation, and the cells are removed via the membrane process. Subsequently, an enzymatic reaction between OPS and sulfur source generates equimolar amounts of L-cysteine and phosphoric acid. The enzyme reaction mixture undergoes the mem-brane process to remove enzymes and residual cells. The L-cysteine is then discharged as the extract through the simulated moving bed process, while phosphoric acid is discharged as the raffinate. The extract undergoes hydrochloric acid addition, followed by evaporative crystalliza-tion, crystal separation, and drying for the production of L-cysteine HCl monohydrate product. The raffinate undergoes ammonia treatment, evaporative crystallization, and crystal separation for the production of sodium ammonium phosphate crystal. Sodium ammonium phosphate can be reused in the OPS fermentation. The amount of sodium added in the previous step affects the sodium content in the sodium ammonium phosphate crystal.”
Comment 6: The process flow for phosphoric acid recycling in the production of CYS HCl is an innovative approach. However, more emphasis should be placed on the practical challenges of scaling up this process. A section discussing potential obstacles in industrial applications would be valuable.
Response 6:
As mentioned in the text already, converting NaOH to NH3 is the most significant obstacle. I have added further details on the industrial hurdles related to this issue as follows (Line 168–175):
“The use of NH3 may increase the vaporization tendency of the sulfur source, so the physical properties, hazards, and safety of the new electrolyte system need to be reassessed. Ammonium ions may impact the enzyme conversion process, requiring process optimization and potentially the development of new enzymes. Furthermore, due to changes in the enzyme conversion reactant, the processes for L-cysteine and phosphate salt crystal production may need to be redesigned. The increased use of ammonia is likely to improve the recovery and reuse rate of phosphoric acid, which may require new design consideration for the fermentation process conditions and strains.”
Comment 7: The paper addresses the environmental benefits of the circular economy model. However, it would be useful to include a more detailed life cycle assessment (LCA) to quantify these environmental benefits, such as reductions in CO2 emissions or water usage.
Response 7:
You are correct. In fact, I have expertise in the LCA process of amino acids, including by-products. Ref. 6 ( https://doi.org/10.1016/j.jclepro.2024.142700 ) is an example of the LCA results for the L-methionine process that I conducted. However, conducting such LCA work involves considering many scenarios, which makes it quite complex and time-consuming. As a result, I could not include the LCA results in this opinion article.
Comment 8: The analysis provided in Table 2 is comprehensive, especially in terms of raw material and energy savings. However, the economic impact of different market conditions (such as fluctuating raw material prices) could be explored to make the analysis more robust and applicable to different regions or industries.
Response 8:
Although incorporating the effects of material prices over time and across locations could make the analysis more robust, it does not have a significant enough impact to alter the conclusion of this opinion article. Therefore, a specific sensitivity test for prices was not conducted.
Instead, the data I provided is based on average values from 2023 data in actual production sites. I have added the following details to the captions of Table 1 and 2 (Line 94–95 and 137–138).
“Table 1. Economic impact analysis of AMS reuse in LYS HCl production. The price information for NH3 ,H2SO4, and steam is based on the average local prices in Indonesia for 2023.”
“Table 2. Economic impact analysis of acetic acid reuse in MET production. The price information for sugar and steam is based on the average local prices in Malaysia for 2023.”
Comment 9: The paper outlines the economic advantages of recycling electrolyte residues, but a deeper discussion of the long-term sustainability of such systems, including potential regulatory challenges or future trends in waste management regulations, would add value to the argument.
Response 9:
I would like this article to focus more on the actual plant-scale (middle-term size and time scale). As an engineer who creates actual economic value in the amino acid industry, I wanted to emphasize process optimization and technological innovation to minimize waste, rather than long-term actions.
Comment 10: While the paper mentions that further research is required, such as increasing ammonium ion content in phosphoric acid salts, a more detailed roadmap or timeline for such research would make the paper’s conclusion more actionable. Highlighting specific areas where breakthroughs are needed (e.g., material innovations or process engineering) would guide future work in the field.
Response 10:
My company has a specific roadmap and timeline, but due to confidentiality, I was unable to disclose them. Instead, as mentioned in Response 6, I have outlined the necessary research topics. (Linx 168–175)

Round 2
Reviewer 1 Report
Comments and Suggestions for Authors
Plant-Scale Circular Economy Using Biological Reuse of Electrolyte Residues in Amino Acid Industry
The authors have revised the manuscript, and the revised manuscript may be considered for publication.
From the previous comments.
It seems there may have been some ambiguity in the question. Could the author please provide further clarification? Specifically, it would be helpful for the readers to understand what characteristics make the electrolyte residues from the amino acid industry effective as fertiliser. What distinguishes these residues and makes them particularly novel for use as fertiliser?
The authors have revised the manuscript, and the revised manuscript may be considered for publication.
Reviewer 2 Report
Comments and Suggestions for Authors
It could be accepted in its current form.